# The Effective Field Theory of Dark Energy Diagnostic of Linear Horndeski Theories After GW170817 and GRB170817A

**Louis Perenon** [1,2,]*  **and Hermano Velten** [3,]*

1   Department of Physics & Astronomy, University of the Western Cape, Cape Town 7535, South Africa
2   Cosmology and Gravity Group, Department of Mathematics and Applied Mathematics, University of Cape Town, Rondebosch, Cape Town 7701, South Africa
3   Departamento de Física, Universidade Federal de Ouro Preto, Ouro Preto MG 35400-000, Brazil
*   Correspondence: lperenon@uwc.ac.za (L.P.); hermano.velten@ufop.edu.br (H.V.)

**Abstract:** We summarize the effective field theory of dark energy construction to explore observable predictions of linear Horndeski theories. We review the diagnostic of these theories on the correlation of the large-scale structure phenomenological functions: the effective Newton constant, the light deflection parameter, and the growth function of matter perturbations. We take this opportunity to discuss the evolution of the bounds the propagation speed of gravitational waves has undergone and use the most restrictive one to update the diagnostic.

**Keywords:** dark energy; modified gravity; large-scale structures

## 1. Introduction

Models incorporating an extra scalar degree of freedom to the Einstein–Hilbert action to explain the accelerated phases of the universe's expansion are numerous, and one is tempted to ask whether a way to describe all these theories in a common framework exists. This unifying description should grant the user the possibility to study and test many models against observations at once. A promising way to achieve this goal is to use *effective field theory*. Effective field theory is a description of the low energy scales of a more fundamental theory. The advantage of using such a description is that one only deals with the degrees of freedom associated with the low energy part of the theory, thus integrating out higher energy scales sometimes irrelevant for the problem at hand.

In the context of attempting an explanation of cosmic acceleration thanks to either Dark Energy (DE) or Modified Gravity (MG), one is generally concerned by the cosmological evolution of the universe. The expansion of the universe is a low energy process, since the energy scale associated with DE, $m_{\rm de}$, can be estimated from the first Friedmann equation by

$$H^2 = \frac{m_{\rm de}^4}{M_{\rm pl}^2} \rightarrow m_{\rm de} = \sqrt{M_{\rm pl}\, H_0} \sim 1 \text{ meV}. \tag{1}$$

It follows that short scale and high energy interactions are often not relevant when comparing DE models to cosmological observations; hence (1) places DE as a suitable ground for an effective field theory description. This has been proposed in the recent years: the *effective field theory of dark energy*, EFT of DE hereafter. Such a common description has enabled a large amount of studies on the theoretical side and observational side. Notably, using the EFT of DE allows one to derive observational constraints, see for instance [1–4], and to explore straightforwardly the observable predictions of theories, see for example [5–7].

Aside of obtaining observable constraints within a theoretical framework, MG has also undergone model independent constraints. For example, the detection of gravitational waves produced by the merging of two neutron stars by the LIGO/VIRGO collaboration, GW170817, in conjunction with the detection of its electromagnetic counterpart, GRB170817A, detected by the FERMI satellite [8,9] impose the speed of gravitational waves to be bounded extremely close to that of light at low redshifts. This could therefore be a stringent constraint for MG and hence models adding one extra scalar degree of freedom to General Relativity (GR) [10–15]. See [16] for a review and [17] for more details on DE in light of multi-messenger astronomy. However, the implications for scalar-tensor theories are yet not fully assessed and are still prone to debate [18]. The purpose of this contribution is the following. We start by concentrating on linear Horndeski theories described by the EFT of DE [1,19,20] and we give a brief review of its construction in Section 2. Horndeski theories [21] are the most general 4-dimensional scalar-tensor theories which retain the field equations at most second order. Later on, in Section 3, based on [5], we use the EFT of DE formulation to present the predictions of linear Horndeski theories, by setting the speed of gravitational waves equal to that of light only today, on the correlation of large-scale structure (LSS) observables, such as the effective Newton constant $\mu$, the light deflection parameter $\Sigma$ and the growth function $f\sigma_8$. In Section 4, we discuss the historical evolution of the bounds on the speed of gravitational waves and the implications for Horndeski theories. We also update the diagnostic the correlation of LSS observables produce on linear Horndeski theories, this time, with the speed of gravitational waves equal to that of light at all times.

## 2. Overview of the EFT of DE Construction

Scalar fields are not used exclusively to model late-time cosmic acceleration. Their use is required to break de Sitter invariance in inflation and to drive the inflationary dynamics via the inflaton field, for example. A common description of single scalar field inflationary models is the *Effective Field Theory of Inflation*. It has been initiated in [22] and systematically developed in [23]. This is the seed of the description of DE with an EFT framework. The key point used to produce such a unifying description is to apply an effective field theory construction to cosmological perturbations directly. They are treated as the Goldstone boson of spontaneously broken time translations as we are going to see. Such a description was then used to describe quintessence [24] and later extended to include Horndeski theories in [19,20,25] and [26,27]. The EFT of DE is a linear description as we will see further on, thereby, it does not include non-linearities such as the ones producing screening mechanisms. Nonetheless, at linear level, the EFT of DE can virtually describe all DE or MG theories which include a single scalar degree of freedom in addition to standard gravity. Let us describe in this section upon what foundations the EFT of DE is based, and how it is constructed.

### 2.1. Spontaneous Symmetry Breaking in Cosmology

GR can be seen as a gauge theory thanks to its invariance under general coordinate transformations where it is the metric field that plays the role of the gauge field. In the case of Minkowski or de Sitter spacetimes, time translations are a global symmetry as they contain a time-like killing vector. Hence, one can say that time translations are broken by any spacetimes not bearing such a killing vector. For example, inflation must be quasi-de Sitter from its almost scale invariant primordial power spectrum and the necessary condition to be able to leave the accelerating phase later. Therefore, inflation *spontaneously* breaks time translations and is accompanied by a Goldstone excitation thus. The Goldstone excitation appears upon the application of the Stückelberg mechanism which we will describe further on. Importantly, this phenomenon makes the presence of a scalar field the inevitable consequence of the broken time translation; the basis of an effective field theory of cosmological perturbations [25].

In a cosmological context, a Friedmann-Lemaître-Robertson-Walker (FLRW) background that is neither Minkowski nor de Sitter must yield a propagating scalar degree of freedom. These scalar fluctuations are the *adiabatic perturbations* in the case of Inflation. Moving to DE makes the description

a little more subtle since matter fields must be involved. The way to corner this difficulty is to apply the EFT construction solely to the gravitational sector, thereby assuming the Weak Equivalence Principle to be valid and thus considering the matter fields to couple universally to the metric through the standard covariant matter action. We are therefore considering the existence of a *Jordan metric*. More complicated set-ups have been explored in [28–30] for example.

## 2.2. Unitary Gauge and the Action

Before presenting the EFT of DE action we must define the gauge in which it is produced: the unitary gauge. We follow the presentation given in [19,20,25] using the redefinition of the coupling function proposed in [1]. The unitary gauge corresponds to the choice of basis in which the Goldstone boson components of a field responsible for the spontaneous symmetry breaking disappear.

In a perturbed FLRW universe, the extra scalar degree of freedom should be decomposed as $\phi(t, \vec{x}) = \bar{\phi}(t) + \delta\phi(t, \vec{x})$. Then, the crucial simplifying step is to choose the time coordinate to be function of $\phi$ such that $\delta\phi = 0$. Doing so, $\phi$ defines a preferred time slicing ($\phi = const.$) and constant time hypersurfaces coincide with constant scalar field hypersurfaces. The action will hence not bear the scalar field and it is built with the unit vector $n_\mu$ defined perpendicular to the time slicing:

$$n_\mu = -\frac{\partial_\mu \phi}{\sqrt{-(\partial_\mu \phi)^2}} \rightarrow -\frac{\delta_\mu^0}{\sqrt{-g^{00}}} \; . \tag{2}$$

This construction implies that the EFT of DE action will include 4-d covariant terms such as the Ricci scalar, any curvature invariant, any contractions of tensors with $n_\mu$ and covariant derivatives of $n_\mu$. For the latter, one uses their projection orthogonal to the constant time hypersurfaces, i.e., the extrinsic curvature tensor

$$K_{\mu\nu} = h_\mu^{\;\sigma} \nabla_\sigma n_\nu \; , \tag{3}$$

with the induced metric defined as $h_{\mu\nu} = g_{\mu\nu} + n_\mu n_\nu$ and $n^\sigma \nabla_\sigma n_\nu \propto h_\nu^{\;\mu} \partial_\mu g^{00}$. As a result, the EFT of DE action of Horndeski theories is

$$S = \int d^4x \sqrt{-g} \frac{M^2(t)}{2} \left[ R - \lambda(t) - \mathcal{C}(t) g^{00} + \mu_2^2(t) \left(\delta g^{00}\right)^2 - \mu_3(t) \, \delta K \delta g^{00} \right.$$
$$\left. - \epsilon_4(t) \left(\delta K^2 - \delta K_{\mu\nu} \delta K^{\mu\nu}\right) + \mathcal{L}_{\mathrm{m}}(g_{\mu\nu}, \psi) \right], \tag{4}$$

where $(M^2, \lambda, \mathcal{C}, \mu_2^2, \mu_3, \epsilon_4)$ are the so-called coupling functions, i.e., the structural functions of time that scale the evolution of the background and perturbations. These coefficients are indeed made time dependent since time translations are broken and they are organized in the order of perturbations, i.e., operators beyond the linear order do not affect the background evolution of the universe, hence $(\mu_2^2, \mu_3, \epsilon_4)$ scale only the perturbations. We refer the reader to [19,20] for example for more involved actions in the EFT of DE form.

## 2.3. Stückelberg Mechanism and Stability of Theories

The method to make the Goldstone excitation appear in the EFT of DE action (4) is the Stückelberg mechanism: the broken gauge transformation on the fields in the Lagrangian is forced back by using the time coordinate transformation

$$t \rightarrow \tilde{t} = t + \pi(x^\mu) \; , \tag{5}$$
$$x^i \rightarrow \tilde{x}^i = x^i \; , \tag{6}$$

where $\pi$ is the Goldstone field. From this, time dependent functions in the action will transform as

$$f(t) \rightarrow f(t + \pi(x)) = f(t) + \dot{f}(t)\pi(x) + \dots , \tag{7}$$

while scalars and the volume element will not. Furthermore, since we take the matter action to be covariant and universally coupled to the Jordan metric it will also not transform while curvature invariants will.

One of the advantages of the EFT of DE is its capability to give a straightforward assessment of the stability of a model. The stability conditions can be obtained by applying the Stückelberg mechanism to (4) and choosing the Newtonian gauge

$$ds^2 = -(1 + 2\Phi)dt^2 + a^2(1 - 2\Psi)\delta_{ij}dx^i dx^j . \tag{8}$$

The action takes then the form [1,20]

$$S_\pi = \int a^3 M^2 \left[ A\left(\mu_1, \mu_2^2, \mu_3, \epsilon_4\right) \dot{\pi}^2 - B\left(\mu_1, \mu_3, \epsilon_4\right) \frac{(\vec{\nabla}\pi)^2}{a^2} \right] , \tag{9}$$

where the two stability conditions in the scalar sector, the ghost and gradient free condition, expressed in terms of the coupling functions are respectively given by

$$A = (C + 2\mu_2^2)(1 + \epsilon_4) + \frac{3}{4}(\mu_1 - \mu_3)^2 \geq 0 \tag{10}$$

$$B = (C + \frac{\mathring{\mu}_3}{2} - \dot{H}\epsilon_4 + H\mathring{\epsilon}_4)(1 + \epsilon_4) - (\mu_1 - \mu_3)\left(\frac{\mu_1 - \mu_3}{4(1 + \epsilon_4)} - \mu_1 - \mathring{\epsilon}_4\right) \geq 0 , \tag{11}$$

were the definition of $C$ can be found the next section and for clarity we have defined:

$$\mathring{\mu}_3 \equiv \dot{\mu}_3 + \mu_1\mu_3 + H\mu_3 , \tag{12}$$

$$\mathring{\epsilon}_4 \equiv \dot{\epsilon}_4 + \mu_1\epsilon_4 + H\epsilon_4 , \tag{13}$$

the Brans–Dicke [31] coupling as the time variation of the bare Planck mass $M^2$,

$$\mu_1(t) = \frac{d \ln M^2(t)}{dt} . \tag{14}$$

The definition of the sound speed of DE perturbations follows,

$$c_s^2 = \frac{B}{A} , \tag{15}$$

which is the propagation speed of the scalar degree of freedom. For the stability of tensor perturbations, one needs to study the propagation of tensor modes from the action (4). One must consider the spatial metric

$$h_{ij} = a^2(t)e^{2\zeta}\hat{h}_{ij} , \tag{16}$$

where $\det \hat{h} = 1$, $\hat{h}_{ij} = \delta_{ij} + \gamma_{ij} + \frac{1}{2}\gamma_{ik}\gamma_{kj}$, $\gamma_{ij}$ is traceless and divergence-free, i.e., $\gamma_{ii} = \partial_i\gamma_{ij} = 0$. Then, since tensor and scalar modes are decoupled at the linear level, one can simply replace this metric into the action (4), oducing

$$S_\gamma^{(2)} = \int d^4x \, a^3 \frac{M^2}{8} \left[ (1 + \epsilon_4) \dot{\gamma}_{ij}^2 - \frac{1}{a^2}(\partial_k\gamma_{ij})^2 \right] . \tag{17}$$

This implies the ghost and gradient free conditions for tensor modes to be respectively

$$c_T^2 = \frac{c^2}{1 + \epsilon_4} \geq 0 \, , \tag{18}$$

$$M^2 \geq 0 \, . \tag{19}$$

It is important to see that indeed, the propagation speed of tensor modes can be different from that of light in vacuum, i.e., $c_T \neq c$, depending of the value of $\epsilon_4$.

### 2.4. Equations and Observables

Here we want to obtain the characteristic equations the action produces in order be able to compute observable predictions in the following sections. To do so, we vary the first line of the action (4) with respect to the metric to obtain the background equations of motion. For a flat universe, this yields

$$\mathcal{C} = \frac{1}{2} \left( H \mu_1 - \dot{\mu}_1 - \mu_1^2 \right) - \dot{H} - \frac{1}{2M^2} (\rho_{\mathrm{m}} + p_{\mathrm{m}}) \, , \tag{20}$$

$$\lambda = \left( 5H \mu_1 + \dot{\mu}_1 + \mu_1^2 \right) \dot{H} + 3H^2 - \frac{1}{2M^2} (\rho_{\mathrm{m}} - p_{\mathrm{m}}) \, , \tag{21}$$

with $H(t)$ the Hubble rate, $\rho_{\mathrm{m}}$ and $p_{\mathrm{m}}$ respectively the background energy density and pressure of matter. We have used the perfect-fluid assumption. At this point, once the evolution of $\rho_{\mathrm{m}}$ is provided by the definition of $H(t)$, one can see that the freedom of a linear Horndeski model is represented in the EFT of DE by one constant and five functions of time:

$$\left\{ \rho_{\mathrm{m},0}, \ H(t), \ \mu_1(t), \ \mu_2^2(t), \ \mu_3(t), \ \epsilon_4(t) \right\} \, . \tag{22}$$

Please note that the constant $H_0$ does not appear. It simplifies out of the relevant equations, only the ratio $H/H_0$ plays a role in the evolution of perturbations, and $\rho_{m,0}$ must be deduced once $H(t)$ is provided.

To obtain the equations of motion of the perturbations, one applies Stückelberg trick to the action (4), considers a perturbed metric, and goes through a series of variation of the action. While we refer the reader to the literature [20] for the details of this procedure, we will concentrate here on showing the expression of LSS observables one can get from the equations of motion. For Fourier modes larger than the nonlinear limit $k \leq 0.15 \, \mathrm{h} \, \mathrm{Mpc}^{-1}$ but smaller than the DE sound speed $k \gg aH/c_s$, we can assume the quasi-static approximation to be valid, i.e., when the time derivatives of the fields in the equations of motion are neglected in front of spatial ones. We neglect any anisotropic stress already possible in GR also as they would be largely sub-dominant. With these approximations, the entire set of scalar perturbation equations can be simplified into two equations

$$-\frac{k^2}{a^2} \Phi = 4\pi \, G_{\mathrm{eff}}(t) \delta \rho_{\mathrm{m}} \, , \tag{23}$$

$$\eta(t) = \frac{\Psi}{\Phi} \, . \tag{24}$$

where $\delta \rho_{\mathrm{m}}$ is the matter perturbations and the effective Newton constant $G_{\mathrm{eff}}(t)$ and the gravitational slip parameter $\eta(t)$ of a given EFT of DE model will govern how modifications of gravity evolve in time. We have neglected any scale dependence of the observables since we are interested in observations well inside the horizon. The extra scalar field being invoked to produce cosmic acceleration, its mass must be very low, of order Hubble, and therefore no scale dependence is expected at low scales. The gravitational slip parameter and the effective Newton constant, which we normalize with $G_{\mathrm{N}}$, can be expressed in terms on the EFT coupling functions as

$$\mu = \frac{G_{\text{eff}}}{G_N} = \frac{M^2(t_0)(1 + \epsilon_4(t_0)))^2}{M^2(1 + \epsilon_4)^2} \, \frac{2\mathcal{C} + \mathring{\mu}_3 - 2\dot{H}\epsilon_4 + 2H\mathring{\epsilon}_4 + 2(\mu_1 + \mathring{\epsilon}_4)^2}{2\mathcal{C} + \mathring{\mu}_3 - 2\dot{H}\epsilon_4 + 2H\mathring{\epsilon}_4 + 2\frac{(\mu_1 + \mathring{\epsilon}_4)(\mu_1 - \mu_3)}{1 + \epsilon_4} - \frac{(\mu_1 - \mu_3)^2}{2(1 + \epsilon_4)^2}} \,, \tag{25}$$

$$\eta = 1 - \frac{(\mu_1 + \mathring{\epsilon}_4)(\mu_1 + \mu_3 + 2\mathring{\epsilon}_4) - \epsilon_4(2\mathcal{C} + \mathring{\mu}_3 - 2\dot{H}\epsilon_4 + 2H\mathring{\epsilon}_4)}{2\mathcal{C} + \mathring{\mu}_3 - 2\dot{H}\epsilon_4 + 2H\mathring{\epsilon}_4 + 2(\mu_1 + \mathring{\epsilon}_4)^2} \,, \tag{26}$$

where $t_0$ corresponds to today. The observable closely related to lensing observations from its sensitivity to the Weyl potential $\Phi + \Psi$ is the light deflection parameter, $\Sigma$. It can be deduced from the previous quantities as

$$\Sigma = \frac{\mu}{2}\,(1 + \eta)\,. \tag{27}$$

Furthermore, the effective gravitational constant $\mu$, aying a role on the dynamics of the matter field, contributes naturally as a source term for the evolution of the linear density perturbations of matter $\delta_{\text{m}}$ :

$$\ddot{\delta}_{\text{m}} + 2H\dot{\delta}_{\text{m}} - 4\pi G_{\text{eff}}\rho_{\text{m}}\delta_{\text{m}} = 0\,. \tag{28}$$

The study of the growth of structures can be better characterized by the combination of the growth function $f$ and the variance of matter density perturbation smoothed on 8 Mpc scales $\sigma_8$. The growth rate $f$, using $f = d\ln\delta_{\text{m}}/d\ln a$, can be obtained by converting the previous Equation (28) into

$$3\bar{w}(1 - x)x\frac{df}{dx}(x) + f(x)^2 + \left[2 - \frac{3}{2}\,(\bar{w}(1 - x) + 1)\right]f(x) = \frac{3}{2}x\,\mu\,, \tag{29}$$

where $\bar{w}$ is the constant effective DE equation of state parameter which we set to $-1$ from now on. The amplitude of the variance of linear matter fluctuations $\sigma_8(x)$ should be computed by re-scaling a normalizing value today $\sigma_{8,0}$ as follows:

$$\sigma_8(t) = \frac{D_+(t)}{D_+^{\Lambda\text{CDM}}(t_0)}\sigma_{8,0}\,, \tag{30}$$

where $D_+$ is the growing mode of linear matter density perturbations obtained by integrating the growth rate $f$. $D_+^{\Lambda\text{CDM}}$ is the growing mode computed in a $\Lambda$CDM cosmology. This re-scaling procedure allows one to make sure the $\sigma_8(t)$ of the model considered agrees with the normalization imposed by Cosmic Microwave Background (CMB) constraints in the past while its evolution is free to be driven by the MG effects.

## 3. Phenomenology of LSS Observables

We have summarized in the previous section the construction of the EFT of DE and we have seen a straightforward way to compute observables. These observables give the possibility to trace MG effects in the perturbation sector. We must now make use of that and extract predictions. Notably, investigating correlations of these observable must lead to highlighting clear signatures of these theories.

### 3.1. Parameterization, Models and Method

The first step to get observable predictions is to deal with the background expansion. We have not discussed much about $H(t)$ so far, as a matter of fact, it is a free function and we will choose an explicit form. We fix the Hubble rate H(z) as a function of the redshift to

$$\frac{H^2(z)}{H_0^2} = \Omega_{m,0}(1 + z)^3 + 1 - \Omega_{m,0}\,, \tag{31}$$

for simplicity since recent observations constrain the expansion history of the universe tightly to that of a flat $\Lambda$CDM model [32], see [33] for other possibilities in treating $H(t)$ in the EFT of DE. Note that moving away from such a background evolution has been shown to barely affect stability conditions [34]. Hence, one should not expect the results of this analyses to change if a different background evolution is chosen provided it is consistent with current data. Furthermore, since measurements of the perturbed sector of the universe are often released in a $\Lambda$CDM background, it is best to do so as well for the sake of appropriate comparison. We fix today's value of the fractional matter density to $\Omega_{m,0} = 0.315$ according to [32]. The coupling functions are also free functions of the theory and we must now parametrize them acutely. It proves useful to introduce another time variable, $x$, the reduced matter density of the background. It scales as a function of the redshift as

$$x = \frac{\Omega_{m,0}}{\Omega_{m,0} + (1 - \Omega_{m,0})(1+z)^{-3}}, \tag{32}$$

and smoothly evolves from today, $x = \Omega_{m,0}$, to deep in matter domination, $x = 1$. This being said, it was shown in [6] that expanding the EFT coupling functions up to order 2 in $(x - \Omega_{m,0})$ was necessary to explore all the phenomenology of linear Horndeski theories. One is also given the possibility to model several DE scenarios depending on the past asymptotic value of the couplings [5]. In other words, the parametrization

$$\mu_1(x) = H(1-x)\left(p_{11} + p_{12}(x - \Omega_{m,0}) + p_{13}(x - \Omega_{m,0})^2\right), \tag{33}$$

$$\mu_2^2(x) = H^2(1-x)\left(p_{21} + p_{22}(x - \Omega_{m,0}) + p_{23}(x - \Omega_{m,0})^2\right), \tag{34}$$

$$\mu_3(x) = H(1-x)\left(p_{31} + p_{32}(x - \Omega_{m,0}) + p_{33}(x - \Omega_{m,0})^2\right), \tag{35}$$

$$\epsilon_4(x) = (1-x)\left(p_{41} + p_{42}(x - \Omega_{m,0}) + p_{43}(x - \Omega_{m,0})^2\right). \tag{36}$$

ensures the couplings to vanish at early times. The $p_{ij}$ are hence the free parameters parametrizing the time evolution of the couplings. However, since $\mu_1$ and $M^2$ are linked by (14), this parametrization models what was dubbed Early DE (EDE) in [5] since it implies $M^2(x \to 1) \neq M^2_{\rm pl}$. To fully confine effects of DE to late times, i.e., the Late Dark Energy (LDE) scenario, the additional constraint

$$-\frac{1 - \Omega_{m,0}}{6}\left[2\,p_{12} + p_{13}(1 - 3\Omega_{m,0})\right] + \frac{1}{3}\ln\Omega_{m,0}\left[p_{11} - \Omega_{m,0}\,p_{12} + \Omega_{m,0}^2\,p_{13}\right] = 0, \tag{37}$$

must be applied to ensure $M^2(x \to 1) \to M^2_{\rm pl}$. On the other hand, if one wants to leave the coupling free to induce modifications of gravity at early times, as in Early Modified Gravity (EMG), one can parametrize the couplings as

$$\mu_1(x) = H(1-x)\left(p_{11} + p_{12}(x - x_0) + p_{13}(x - x_0)^2\right), \tag{38}$$

$$\mu_2^2(x) = H^2\left(p_{21} + p_{22}(x - x_0) + p_{23}(x - x_0)^2\right), \tag{39}$$

$$\mu_3(x) = H\left(p_{31} + p_{32}(x - x_0) + p_{33}(x - x_0)^2\right), \tag{40}$$

$$\epsilon_4(x) = \left(p_{41} + p_{42}(x - x_0) + p_{43}(x - x_0)^2\right). \tag{41}$$

Please note that a pre-factor $(1-x)$ remains for $\mu_1$ and that is because it cannot be allowed to vanish. A vanishing $\mu_1$ would imply $M^2$ to diverge from Equation (14). To obtain predictions on the LSS observables, we randomly generate the $p_{ij}$ parameters until we have produced $10^4$ viable models, i.e., models that do not bear ghost, nor gradient instabilities, nor super-luminal propagation speeds $c_S$ and $c_T$, of the LDE, EDE, and EMG scenarios. The action (4) being an expansion in perturbations

and the coupling functions divided by their required power of $H$ are expected to be of order 1. Consequently, we chose to randomly generate the coefficients uniformly in the interval $p_{ij} \in [-1, 1]$.

### *3.2. Correlations of LSS Observables*

The protocol previously mentioned produces the results displayed in Figure 1 for linear Horndeski theories with $c_T(t_0) = c$. We set $p_{41} = 0$ to apply this condition. More details of this work can be found in [5], let us summarize the main observations.

A striking fact one can observe is that despite the large functional freedom in the EFT couplings, the correlations of the LSS observables across redshift display bounded and defined trends due to the viability requirements. Let us inspect them in more details. The first row of Figure 1 depicts the results for the LDE scenario where an alternation of weaker and stronger gravity can be seen. Indeed, at very small redshifts all models bear $\mu > 1$ while at intermediate redshifts models populate the region above and below $\mu = 1$, and the $\mu > 1$ tendency is recovered once $z \geq 2$. This characteristic evolution of $\mu$ has noticeable implications on $f\sigma_8$. The amplitude of the later is always lower than that of $\Lambda$CDM once $z \geq 1.5$. This displacement in redshift in between $\mu$ and $f\sigma_8$ is indeed due to the presence of $x$ in the right-hand side of (29), i.e., a specific feature spotted on $\mu$ at a certain redshift will be reflected at a slightly lower redshift for $f\sigma_8$. We also note that once $z \geq 1$, most of the models predict lower growth of structure than $\Lambda$CDM. One can also add that depending on the redshift, certain quadrants in these correlation planes are completely depopulated, one example being $f\sigma_8 < f\sigma_{8,\Lambda CDM}$ and $\Sigma > 0$ for any $z > 1$.

Let us see how much definite features in the prediction of linear Horndeski models remain when more freedom is allowed, i.e., when moving to EDE and EMG scenarios. The first important observation to mention is the fact that allowing early modifications of gravity also alters the late-time predictions of LSS observables. Considering the EDE case leads to obtaining models favoring lower $\mu$ and $\Sigma$ than $\Lambda$CDM and much lower $f\sigma_8$ for the redshifts displayed in Figure 1. On the other hand, the EMG scenario is the only possibility that allows $\mu > 1$ and $\Sigma > 1$ and $f\sigma_8 > f\sigma_{8,\Lambda CDM}$ for $z > 1$. One must scrutinize the expression of $\mu$ further to understand the reasons behind these new features. A little bit of algebra allows one to transform expression (25) into

$$\mu = \left( \frac{M(\Omega_{m,0})\,(1 + \epsilon_4(\Omega_{m,0}))}{M\,(1 + \epsilon_4)} \right)^2 \left[ 1 + \frac{1 + \epsilon_4}{2B} \left( \frac{\mu_1 - \mu_3}{2(1 + \epsilon_4)} - (\mu_1 + \mathring{\epsilon}_4) \right)^2 \right] \tag{42}$$

where the first term on the right-hand side is the bare modifications of the Newton constant by MG. In other words, it is the modification of gravity that remains even in an environment where the scalar field is decoupled from gravity, i.e., a screened environment. On the other hand, the second term corresponds to the fifth force induced by the extra scalar field. One can thus appreciate this term to always be larger than 1 since our viability requirements imply that $B \geq 0$ and $\epsilon_4 \geq -1$ since $c_T^2 > 0$, as it is expected for a healthy massless spin 0 field, i.e., an attractive force. Please note that the above explains why, by definition, no model will be able to exhibit $\mu < 1$ at redshift zero.

It is this distinction between the components that allows us to understand the new features on the EDE and EMG scenario. For the former, the second term on the right-hand side will always go to one in the past since the couplings are designed to vanish. However, we allow now for values of $M^2$ different than $M_{pl}^2$ deep in matter domination and we observe that the stability requirements favor almost exclusively $M(\Omega_{m,0})\,(1 + \epsilon_4(\Omega_{m,0})) < M\,(1 + \epsilon_4)$, i.e., essentially $M^2 > M_{pl}^2$. Hence, this is the term responsible for weaker gravity, despite the scalar field mediating a fifth force. This is what, in turn, implies that the EDE scenario displays much weaker gravity and lower growth than $\Lambda$CDM as compared to the LDE scenario. In addition, since the couplings still vanish, the gravitational slip parameter will go to unity in the past and therefore this new behavior of $\mu$ implies $\Sigma < 1$ will be favored in the past, as one can clearly see in Figure 1. Once the couplings no longer vanish in the past as it is in the EMG case, the second term on the right hand side of (42) no longer goes to one and

neither does the gravitational slip parameter. This is why values of $\mu > 1$ and $\Sigma > 1$ are allowed in the past. One last striking feature we should discuss is the $45°$ correlation between $\mu$ and $\Sigma$ that draws itself for $z \geq 1$ irrespectively of the scenario. Using (27) one can show that

$$\mu - 1 = \left(\frac{2}{1+\eta}\right)(\Sigma - 1) - \frac{\eta - 1}{1+\eta},\qquad(43)$$

hence the correlation holds when $\eta$ remains sufficiently close to one. This is indeed still the case in the EMG scenario. The fact that a sign agreement between $\mu$ and $\Sigma$ must exist for Horndeski theories has been first conjectured in [35] and further justified in [7].

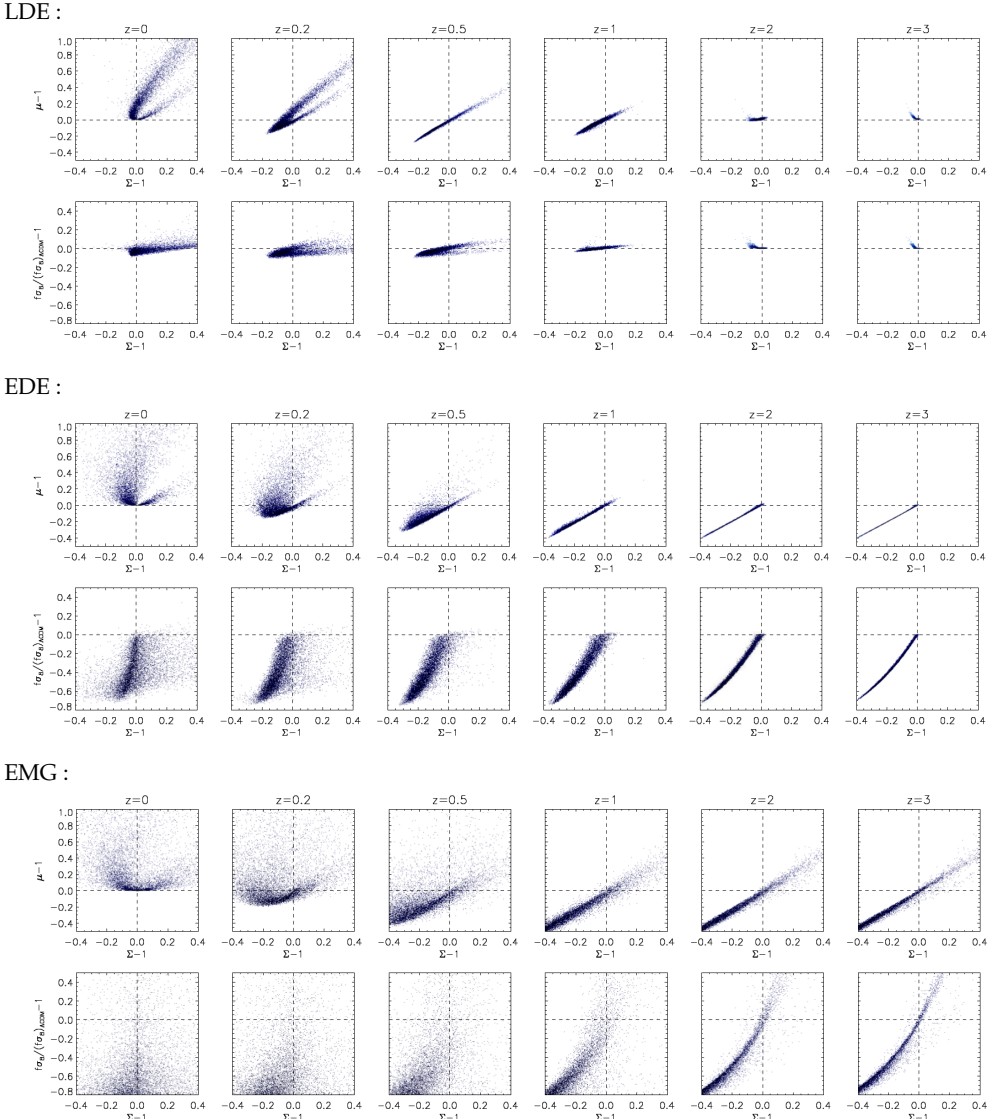

**Figure 1.** We display the correlations $\mu - \Sigma$ and $f\sigma_8 - \Sigma$ for several redshifts $z = \{0, 0.2, 0.5, 1, 2, 3\}$. The first two rows correspond to a generation of $10^4$ viable EFT models in the LDE scenario, the middle two rows $10^4$ models in the EDE scenario and the bottom two rows $10^4$ models in the EMG scenario. All the models bear $c_T(t_0) = c$. In each plot, the $\Lambda$CDM prediction corresponds to the intersection of the two dashed lines and the gray/blue scale highlights the density of points.

## 4. Implications after GW170817 and GRB170817A

Before establishing a diagnostic of linear Horndeski theories from the results of the previous section, we must take into consideration the new bounds on the propagation speed of gravitational

waves (GW). Let us therefore take the opportunity of this section to give a review of the evolution the constraints on $c_T$ have undergone and then dress our diagnostic accordingly.

*4.1. Constraints on the Speed of Gravitational Waves*

As discussed previously, the DE phenomenon, the fact that today's background expansion of the universe is accelerating, could be explained via the idea of modifying Einstein's GR by adding a new degree of freedom, a scalar field. For the particular case of Horndeski theories such a new approach leads to new freedoms and one of them being the speed of propagation of GW $c_T$ to be free. In GR is not the case since the construction of the theory requires that GW travel at the speed of light in vacuum $c_T = c$. If the gravitational framework allows now that $c_T \neq c$, one may wonder what astrophysical and cosmological bounds on $c_T$ exist. Independently of the original physical reason giving rise to such anomalous propagation, one should bear in mind that GW, when analyzed in the context of MG, cannot travel on null geodesics of the background metric as photons do.

One can inquire about the lower bounds on $c_T$, i.e., can GW travel at a speed $c_T < c$. The case $c_T < c$ is very tightly constrained by observations of the highest energy cosmic rays from galactic origin [36]. The idea behind this bound relies on the assumption that if $c_T < c$, there would exist particles moving faster than the speed of gravity and would thereby emit a "gravitational Cherenkov radiation" in a similar analogy to the usual Cherenkov radiation emitted by particles moving faster than light in a medium.

For protons with an energy of $\mathcal{O}(10^{11})$ GeV arriving to Earth from a distance of the order of the galactic center, the bound on the time of flight obtained in [36] induces the bound on the speed of propagation of GW to be

$$\frac{c_T}{c} - 1 \leq 2 \times 10^{-15}. \tag{44}$$

The analysis of the Cherenkov radiation method cannot be used to constrain $c_T > c$ since the gravitational Cherenkov radiation does not exist in this case. It is however worth noting that the typical energy scale involved in such radiated GW tested by the Cherenkov radiation $\sim 10^{11}$ GeV is beyond any reasonable cut-off scale of MG theories designed to explain cosmic acceleration. We also point out to the reader that earlier investigations on the speed of GW in bimetric theories of gravity have been conducted in [37].

Prior to the detection of GW by the LIGO/VIRGO collaboration, the observations of the orbital decay in the PSR B1913+16 binary system, also known as the Hulse-Taylor pulsar, have been used to place stringent upper bounds on $c_T$ of order $\sim 1\%$ [38], the best upper limit on $c_T$ at that time. PSR B1913+16 is a radiating neutron star, i.e., a pulsar, in a binary system with its companion star: another neutron star. This system was the first binary pulsar to be discovered and was awarded the Nobel Prize in Physics on 1993, we cite "for the discovery of a new type of pulsar, a discovery that has opened up new possibilities for the study of gravitation" [1]. This discovery represented the first evidence, although indirect, of the existence of GW. Indeed, the observed orbital decay of this system occurs from the loss of energy in the system which is associated with the emission of GW. The Hulse-Taylor pulsar is also an excellent confirmation of GR predictions. In [38], from the computation of the post-Keplerian parameters in the Hulse-Taylor, the following bound on the speed of GW was set:

$$0.995 \leq \frac{c}{c_T} \leq 1.00. \tag{45}$$

Aside of astrophysical observables as discussed above, one can use cosmological data to place constraints on $c_T$. CMB observations are helpful to distinguish characteristic signatures of GW in its B-mode polarization spectrum. In the latter, two main distinct effects can be found. Firstly, MG and in

---

[1] https://www.nobelprize.org/prizes/uncategorized/the-nobel-prize-in-physics-1993-1993/

some cases its intrinsic anisotropic stress [39] can lead to a lensing contribution to B-modes. In general, a modified lensing potential amounts to effects in the TT, EE, and BB spectra. The other effect, and the most important for us here, is the one appearing if $c_T$ differs from $c$. The position of the peak of the primordial B-modes can shift since $c_T$ sets the time at which they cross the horizon. However, the bounds from CMB B-modes on $c_T$ are less precise $\sim 10\%$ [2,40].

In 2016, the LIGO collaboration announced the directed observation of GW emitted from merging black hole binaries [41] without the electromagnetic counterpart. This, in principle, allows one to infer a bound on $c_T$. However, due to the fact there were only three detections at that time and the high uncertainty in the position of the sources, the data was capable of setting looser bounds: $0.55\,c < c_T < 1.42\,c$ [42].

On August 17th, 2017, the LIGO/VIRGO collaboration triggered the advent of the multi-messenger astronomy era in which both the gravitational and electromagnetic spectrum of an astronomical event had been detected [8,9]. The GW signal (GW170817) was followed by a short gamma ray burst (GRB170817A) only $1.74 \pm 0.05$ s after its arrival [8]. Due to the simultaneous detection by three facilities (the two LIGO detectors and the VIRGO detector) the source was localized at a distance of $40^{+8}_{-14}$ Mpc. The LIGO/VIRGO collaboration used as the reference distance the value 26 Mpc to set the most conservative constraints on $c_T$. By assuming both the GW signal and the gamma ray photon were emitted at the same time it is, therefore, ssible to set an upper limit on $c_T$. On the other hand, to find the lower bound, one assumes the gamma ray was emitted 10 s after the GW which is the maximum delay allowed by current astrophysical models of the collapse. Then, in this case, since the photon arrived only 1.7 s after the GW, this means the GW has travelled slightly slower than light. One is therefore able now to place both an upper and lower bound on $c_T$. Finally, these results from the detection of GW170817 and GRB170817A have improved immensely the bound on $c_T$ by several orders of magnitude:

$$-3 \times 10^{-15} < \frac{c_T}{c} - 1 < 7 \times 10^{-16}. \tag{46}$$

One would be too abrupt in thinking that such a constraint bounds MG theories to necessarily display $c_T = c$ at all times. While it was thought first that this last bound would stringently confine the possible quartic and quintic Lagrangians of Horndeski theories, i.e., the Lagrangians responsible for an anomalous gravitational wave speed, it was later discovered that some models can be effectively "rescued" [43]. Beyond the previous, this stringent bound on $c_T$ must be placed in time and scale. On the one hand, this bound corresponds to low redshifts $z \leq 0.01$. Therefore, it certainly concerns late-time cosmic acceleration but not the early universe. Indeed, $c_T$ could vary across time and be equal to $c$ today without a fine-tuning [44]. On the other hand, the wavelengths associated with these GW and those relevant for cosmic acceleration differ by an order $\mathcal{O}(10^{19})$ [45], thus very close to the cut-off scale of Horndeski theories and many DE models [18]. Moreover, UV completion can also enable an anomalous speed of gravitational waves to equate that of light for the frequencies observed for the GW170817 and GRB170817A event [18]. In light of this, that is why we have chosen to present the results for models with $c_T(t_0) = c$, as it was done in [6], and will, for the sake of generality, also consider the most restrictive choice $c_T(\forall t) = c$ further on.

## 4.2. The Diagnostic of Linear Horndeski Theories

Let us now see what diagnostic on linear Horndeski theories we can produce. The analysis commented in Section 3.2 gives the flavor of how joint measurements of the LSS observables $\eta$, $\Sigma$ and $f\sigma_8$ would lead to strong indications as to whether linear Horndeski theories are favored by data and what type of DE scenario is more viable. Let us therefore use these correlations to foster a diagnostic of linear Horndeski theories which yield $c_T(t_0) = c$. The analysis of how the models populate the correlation of LSS plane allows us to draw the diagram in Figure 2. We conclude that linear Horndeski models in which $c_T(t_0) = c$ is imposed would be ruled out should future observations point to

- $\mu$ and $\Sigma$ of opposite sign for $z > 1.5$,
- $\mu < 1$ at $z = 0$.

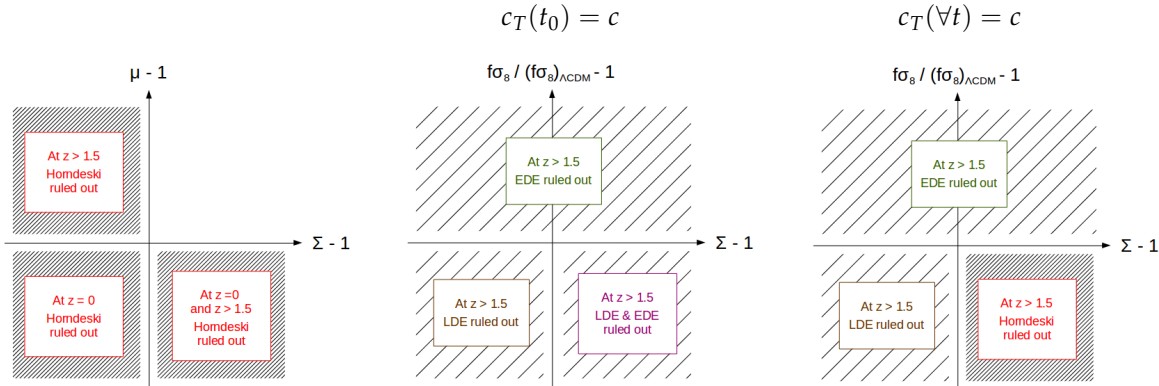

**Figure 2.** Schematic diagrams of the correlations of LSS observable giving the diagnostic of linear Horndeski theories valid for both cases $c_T(t_0) = c$ and $c_T(\forall t) = c$ (left diagram). The diagnostic of the DE scenario embedded within linear Horndeski theories is shown for the case $c_T(t_0) = c$ (middle diagram) and for the case $c_T(\forall t) = c$ (right diagram).

The $f\sigma_8$ and $\Sigma$ plane will allow discrimination between the DE scenario embedded within linear Horndeski theories once higher redshift ($z \geq 1.5$) measurements become available. Indeed, we conclude for the case $c_T(t_0) = c$ that

- the LDE case should be ruled out if $f\sigma_8 < (f\sigma_8)_{\Lambda CDM}$ at $z > 1.5$,
- the EDE case should be ruled out if $f\sigma_8 > (f\sigma_8)_{\Lambda CDM}$ at $z > 1.5$ or $f\sigma_8 > (f\sigma_8)_{\Lambda CDM}$ and $\Sigma > 1$ at $z > 1.5$.

To finish let us add that this diagnostic was tested for robustness in [5], where neither relaxing the viability on conditions on the propagation speed of perturbations, nor changing the constant effective DE equation to $\bar{w} = -1.1$ or $\bar{w} = -0.9$, nor randomly generating the parameters from a Gaussian distribution altered the diagnostic.

It is now important we further update our diagnostic given the stringent bound on $c_T$ we discussed in the previous section. For the sake of completeness, we now test the most restrictive possibility and set $\epsilon_4 = 0$, i.e., $p_{41} = p_{42} = p_{43} = 0$, so as to have $c_T = c$ at all times. Following the same protocol as exposed in 3.1 we obtain the results displayed in Figure 3. The main consequence of this new condition is to tighten the dispersion of the points in the correlation space, i.e., the features in the latter are now even sharper than in Figure 1. For instance, once $z \geq 2$ the LSS observables $\mu$, $\eta$, $\Sigma$ and $f\sigma_8$ are really confined to be that of the predictions from the standard model in the LDE scenario. With this new condition on $c_T$, the expressions of the observables simplify significantly. For instance, the effective Newton constant reduces to

$$\mu = \left(\frac{M(\Omega_{m,0})}{M}\right)^2 \left[1 + \frac{(\mu_1 + \mu_3)^2}{4B}\right] , \tag{47}$$

where now

$$B = C + \frac{\mathring{\mu}_3}{2} + \frac{1}{4}(\mu_1 - \mu_3)(3\mu_1 + \mu_3) \geq 0 . \tag{48}$$

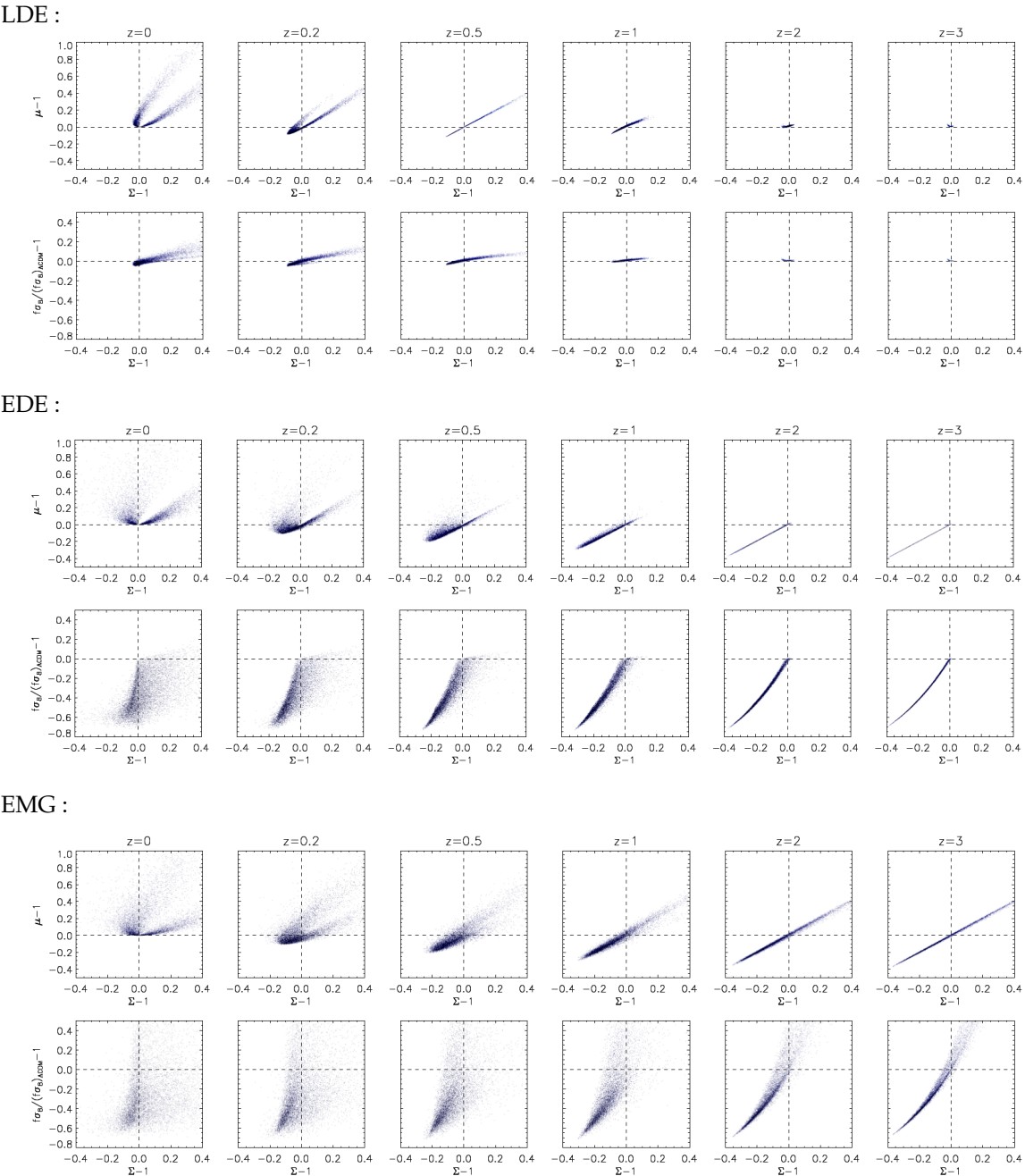

**Figure 3.** We display the same correlations as in Figure 1 however here all the models bear $\epsilon_4 = 0$ hence $c_T = c$, at all times.

Therefore, one can see that the weakening of gravity in linear Horndeski theories, hence the lower growth of structure predictions, are now solely due to the bare Planck mass $M^2$, i.e., the Brans–Dicke coupling $\mu_1$ equivalently, since $\epsilon_4$ has vanished.

A new feature appearing in Figure 3 crucial to note is the following. For redshifts $z \geq 1.5$ the $45°$ correlation between $\mu$ and $\Sigma$ whether the EDE or EMG scenario are considered is now sharply highlighted. The models span a tight thin line in the correlation plane. Consequently, it stands out clearly now that no model, irrespectively of the DE scenario, lies in the quadrant $f\sigma_8 < f\sigma_{8,\Lambda\text{CDM}}$ and $\Sigma > 0$. In conclusion, an important upgrade of the diagnostic can be made as depicted in Figure 2. While the $\mu - \Sigma$ plane bears no change with respect to the case $c_T(t_0) = c$, the bottom right quadrant of

the $f\sigma_8 - \Sigma$ plane now no longer only allows discarding of the LDE and EDE scenarios but the whole class of linear Horndeski theories.

## 5. Conclusions

In this contribution, we have presented a unifying framework: the effective field theory of DE and used it in the context of recent stringent constraints on MG such as the bound on the speed of propagation of GW. The reason for this choice of framework stems from the fundamental link the EFT of DE has with scalar-tensor theories. The coupling functions of its action characterize the linear evolution of matter perturbations in the universe and parametrize these theories in terms of structural functions of time which, in turn, appear naturally in the expression of observables. This leads to an easy comparison of theoretical predictions with observations. One profound revelation of the EFT of DE description is the presence of a scalar field arising as the inevitable consequence of the spontaneously broken time translations of spacetime. This description has its advantages and drawbacks. Being a linear description, it cannot describe nonlinear regimes such as the screening mechanism. Moreover, the unitary gauge description loses the apparent covariance of the theory; however, it has the benefit of classifying operators in order of perturbations. Nevertheless, the covariant description of the theory can be recovered thanks to the Stückelberg trick mechanism.

With this common formulation, edictions of linear Horndeski theories can be straightforwardly computed. We find that studying the correlations of LSS observables, the effective Newton constant, the light deflection parameter, and the growth function, to be key in discriminating parts of the theories in light of future surveys. Notably, it is interesting to compile our results into a diagnostic of linear Horndeski theories. It essentially tells us that depending where future measurements will point in the correlation's plains, certain parts or the hole theory could be ruled out. We also found this diagnostic to become more stringent once the speed of GW is fixed to that of light at all times. In light of this contribution, generalizations of this diagnostic should be explored. Indeed, we have focused on scales much smaller than the Hubble radius; however, as data improves on larger scales, our diagnostic should be extended to include possible scale dependence coming from Hubble scale effects [46]. Furthermore, it would be interesting to assess to what level our diagnostic holds when more involved scenarios are taken into account. Within the EFT of DE, one could notably study GLPV theories, the so-called Beyond Horndeski theories [47,48], interacting dark energy models [28], models exhibiting kinetic matter mixing [30] or even the new larger branch of higher-order scalar-tensor theories [49].

As a final concluding remark of this contribution, let us point out that treating broken symmetries, as the EFT of DE does, goes indeed beyond high energy physics. For instance, some of the authors that developed the EFT of DE have applied the EFT method to condense matter physics [50]. This enabled the description of *framids*, a common description of matter states, and the prediction of new hypothetical states.

**Author Contributions:** Writing, L.P. and H.V.

**Funding:** L.P. acknowledges financial support from the National Research Foundation (South Africa) (Grant No. 119652). H.V. is partially funded by CNPq (Brazil) under contract 422360/2018-0.

**Conflicts of Interest:** The authors declare no conflict of interest.

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
