# Peer review of "The Effective Field Theory of Dark Energy Diagnostic of Linear Horndeski Theories After GW170817 and GRB170817A"

_universe, doi:10.3390/universe5060138_

Round 1
Reviewer 1 Report
The authors give a very clear introduction on the effective field theories technique applied to gravity and in particular to the case of gravity with an extra scalar degree of freedom. Then, they build up the new cosmological observables that this kind of theory predicts, also in a very clear manner. Finally, they compare their results with observation, especially in light of the recent very powerful constraint on the speed of gravitational waves. The manuscript is very clearly written and scientifically sound, therefore I recommend it for publication.
Author Response
We would like to warmly thank the referee for dedicating time to our manuscript and reviewing it.
Reviewer 2 Report
The cosmological observables and the speed of gravitational waves in the linear Horndeski theory are discussed in this paper. The investigations at the cosmological observables are valuable.
I am glad to recommend this paper after three minor corrections.
The evolution of the Hubble parameter is shown in Eq. (31), in which the authors take a constant value for the dark energy density. However, a time dependent energy density can have a non-negligible effect to the LSS spectrum. Thus, the authors have to justify their reason to use the LCDM background.
The authors show the asymptotic behaviour in the sentence around Eq. (37) that $M(x -> 1) -> M_{pl}$. This point is very important, and the behaviour must be applicable to the high redshift regime.Is the argument still valid when the energy density of the relativistic fluid is taken into account?
The cosmological observables are discussed in Fig. 1-3 with $ -1 < p_{ij} < 1 $. I can not find the relation between $p_{ij}$ and the model parameters in the action. Showing the explicit formula would be helpful to the reader. In addition, it seems $p_{ij}$, in general, can be arbitrary, so a brief justification to the fitting region, $-1 < p_{ij} < 1$, is needed.
Author Response
We would like to thank the referee for dedicating time to our manuscript and for his/her comments. We reply to the points raised by the referee hereafter.
Point 1
The evolution of the Hubble parameter is shown in Eq. (31), in which the authors take a constant value for the dark energy density. However, a time dependent energy density can have a non-negligible effect to the LSS spectrum. Thus, the authors have to justify their reason to use the LCDM background.
Indeed a varying dark energy density can have an effect on the observations of the perturbed sector. This is however beyond the scope of this proceeding and beyond the scope of the paper and presentation it is based on. Our justification for this choice can be found in the first paragraph of Section 3.1. This justification is based on the fact that from the latest observations and notably the constraints established by the Planck collaboration, LambdaCDM remains the best model to fit, at least, the background evolution of the universe. We do point the reader to the literature for cases where the evolution of H(t) is not that of LCDM. We also justify this choice further by the sentence, we quote : "Furthermore, since measurements of the perturbed sector of the Universe are often released in a $\Lambda$CDM background, it is best to do so as well for the sake of appropriate comparison." We now include a new reference [Brando et al] which shows that the stability conditions in such theories barely change in a modified background and a sentence in bold font explaining this further. In addition, one must note however that we explain in Section 4.2, we quote: " To finish let us add that this diagnostic was tested for robustness in [1], where neither relaxing the viability on conditions on the propagation speed of perturbations, nor changing the constant effective dark energy equation to w̄ = − 1.1 or w̄ = − 0.9, nor randomly generating the parameters from a Gaussian distribution altered the diagnostic." This indeed points out that constant equation of state parameters different than LambdaCDM do not change our results. We believe therefore that this gives us a hint that choosing a varying equation of state but remaining within bounds allowed by observations would virtually not affect our conclusions. This remains yet to be thoroughly investigated in future work indeed.
Point 2
The authors show the asymptotic behaviour in the sentence around Eq. (37) that $M(x -> 1) -> M_{pl}$. This point is very important, and the behaviour must be applicable to the high redshift regime.Is the argument still valid when the energy density of the relativistic fluid is taken into account?
In terms of the modelling of the coupling functions of the EFT action (4). Both LDE and EDE cases are modelled through the parameterisation (33-36). Yet to confine the dark energy to late times, the condition (37) must be applied to (33) and this characterises the LDE scenario. If the condition is not applied, one accesses the EDE scenario instead. In other words, the results for LDE correspond to models where $M(x -> 1) -> M_{pl}$ while EDE $M(x -> 1) -> const. $. This is, however, a description that remains within DE and matter domination as x->1 signals the deep matter domination limit. We know from observations that the universe at early times should be very close to GR (CMB, BBN etc) and requiring from a theoretical point of view $M(x -> 1) -> M_{pl}$ seems a fair assumption. From the data constraining point of view, $M(x -> 1) -> const. $. should be the way to go where the data would constrain any deviations, and most probably impose bounds very close to $M(x -> 1) -> M_{pl}$. In this manuscript, we are not considering data and therefore we wanted to show both approaches. We believe that this discussion still holds whether or not the relativistic fluid is taken into account. Adding the latter would benefit greatly a data analyses as in would allow the application of early time probes and yield constraints on the deviation $M(x -> 1) -> const. $.
Point 3
The cosmological observables are discussed in Fig. 1-3 with $ -1 < p_{ij} < 1 $. I can not find the relation between $p_{ij}$ and the model parameters in the action. Showing the explicit formula would be helpful to the reader. In addition, it seems $p_{ij}$, in general, can be arbitrary, so a brief justification to the fitting region, $-1 < p_{ij} < 1$, is needed.
We have some difficulties to fully understand this comment from the referee. We briefly explain again our reasoning. This might guide the referee through our arguments. The section 3.1 is dedicated to the thorough presentation of how we parametrise the models. We show explicitly how the p_{ij} are the models parameters and how they relate to the coupling functions of the action. In addition, at the end of the aforementioned section, we also justify the choice $-1 < p_{ij} < 1$ with the explanation, we quote: "The action (4) being an expansion in perturbations, the coupling functions divided by their required power of H are expected to be of order 1. Consequently, we chose to randomly generate the coefficients uniformly in the interval $-1 < p_{ij} < 1$."
Reviewer 3 Report
Referee Report on the manuscript universe-508081
This is my report on the article “The effective field theory of dark energy diagnostic of linear Horndeski theories after GW170817 and GRB170817A”, by L. Perenon and H. Velten.
In this article, the authors discuss on the linear Horndeski theories in view of the effective field theory of dark energy (EFT of DE), i.e., in the context of the low-energy scale of a more fundamental theory. Horndeski theories are the most general four-dimensional scalar-tensor theories that retain the associated field equations to second order. The authors use the EFT of DE formulation to demonstrate the predictions of linear Horndeski theories on the large-scale structure observables, such as the effective Newton constant, the light deflection parameter, and the growth function. After discussing on the bounds of the speed of gravitational waves and the associated implications for Horndeski theories, they update the diagnostic that the correlation of large-scale structure observables produces on linear Horndeski theories, admitting that the speed of the gravitational waves equals to the speed of light, at all times.
I believe that this is a competent work, in which, the major large-scale structure diagnostics on linear Horndeski theories are discussed and scrutinized in the light of the GW170817 and GRB170817A data.
The text is, generally-speaking, clear and well-written, while I found (and read) most of the (well-posed set of) References. I believe that, the authors have done a good job.
For all these reasons, I recommend the article “The effective field theory of dark energy diagnostic of linear Horndeski theories after GW170817 and GRB170817A”, by L. Perenon and H. Velten, to be published in Universe.

Author Response
We would like to warmly thank the referee for dedicating time to our manuscript and pointing out an English mistake.
We have corrected the sentence suggested by the referee. This can be found in bold font in the last paragraph of Section 1 of the updated version.